# Whole Exome Sequencing for the Identification of Mutations in Bone Marrow CD34+Cells in Hodgkin Lymphoma

**DOI:** 10.3390/cimb47110880

**Published:** 2025-10-23

**Authors:** Phan Thi Hoai Trang, Do Thi Trang, Pham Thi Huong, Pham Viet Nhat, Mentor Sopjani, Nguyen Hoang Giang, Nguyen Xuan Canh, Nguyen Van Giang, Nguyen Trung Nam, Nguyen Ba Vuong, Vu Duc Binh, Nguyen Thi Xuan

**Affiliations:** 1103 Military Hospital, Vietnam Military Medical University, 261 Phung Hung, Ha Dong, Hanoi 100803, Vietnambavuongsang@gmail.com (N.B.V.); 2Department of Pathophysiology, Vietnam Military Medical University, 261 Phung Hung, Ha Dong, Hanoi 100803, Vietnam; 3Institute of Biology, Vietnam Academy of Science and Technology, 18 Hoang Quoc Viet, Cau Giay, Hanoi 100722, Vietnam; dothitrang23021993@gmail.com (D.T.T.); huongthipham48@gmail.com (P.T.H.); phamvietnhat.pvn@gmail.com (P.V.N.); hoanggiang6891@gmail.com (N.H.G.); nam@ibt.ac.vn (N.T.N.); 4Publishing House for Science Technology, Vietnam Academy of Science and Technology, 18 Hoang Quoc Viet, Cau Giay, Hanoi 100722, Vietnam; 5Faculty of Medicine, University of Prishtina, 10130 Prishtinë, Kosovo; mentor.sopjani@uni-pr.edu; 6Faculty of Biotechnology, Vietnam National University of Agriculture, Gia Lam, Hanoi 100800, Vietnam; xuancanh79@yahoo.com (N.X.C.); vangianghua@gmail.com (N.V.G.); 7National Institute of Hematology and Blood Transfusion, Pham Van Bach, Hanoi 100400, Vietnam

**Keywords:** *CNN2*, CD34+ cells, Hodgkin lymphoma, *MUC4*, whole-exome sequencing

## Abstract

Background: Classical Hodgkin lymphoma (cHL) is a rare B-cell malignant neoplasm, characterized by the presence of rare mononucleated Hodgkin and multinucleated Reed–Sternberg cells (HRS). CD34+ cells are highly expressed on lymphoma stem cells in bone marrow (BM). Little is known about gene mutations in BM CD34+ cells of cHL. In this study, whole exome sequencing (WES) was performed and high-frequency mutation genes were examined through their expression levels. Materials and Methods: The influence of the variants on protein function was predicted with in silico tools or public databases. Gene expression levels were determined by quantitative real-time PCR. Results: WES assay from BM CD34+ cells in thirty cHL patients revealed that three variants were detected in known cHL-associated genes, including NCF1 (13.33%), MMP9 (3.33%), and VDR (3.33%). We also observed other candidate genes including CNN2 rs77830704 (76.67%), CNN2 rs78386506 (63.33%), MUC4 p.Y3278_Q3209Del (66.67%), MUC4 p.P1076_P1124Del (33.33%), MUC4 rs748236754 (26.67%), MUC4 p.P1609Ins (23.33%), MUC4 rs748705487 (20%), MUC4 p.P4121_P4137Del (16.67%), MTSS2 rs531163149 (13.33%), KMT2C rs201834857 (20%), HAVCR2 rs184868814 (16.67%), and TCF19 rs541001159 (13.33%). Moreover, the low levels of MUC4 were associated with an increase in neutrophil-to-lymphocyte ratio and the low CNN2 expression group had higher levels of LDH, suggesting that the low expressions of CNN2 and MUC4 might be important risk factors for poor prognosis in cHL. Conclusions: WES revealed significantly mutated genes, most of which were associated with the physiological activation of lymphoma cells. This finding contributed to the identification of novel gene variants that might impact on the function of BM CD34+ cells in cHL patients.

## 1. Introduction

Classical Hodgkin’s lymphoma (cHL) is a rare B-cell malignant neoplasm, characterized by the emergence of giant abnormal cells called mononucleated Hodgkin and multinucleated Reed–Sternberg cells (HRS) that originate from germinal center B-cells. The tumor cells comprise <2% of the inflammatory tumor microenvironment, including T helper cells, natural killer (NK) cells, dendritic cells, tumor-associated macrophages (TAMs), eosinophils, stromal cells, plasma cells, and histiocytes [1]. The uncontrolled activation of HRS cells and/or the surrounding tissue leads to a defective inflammatory response, characterized by the abnormal expression of multiple cytokines such as TGF-β [2]. The HRS cells show constitutive NF-κB activation and the overexpression of CD30 as well as a lack in expression of most B-cell markers, such as CD19, CD20, and CD79a [3]. Low B-cell counts are associated with risk of progression and relapse in cHL [4]. In the World Health Organization (WHO) lymphoma classification, cHL has been categorized into four different histological subtypes: nodular sclerosis (NSHL); mixed cellularity (MCHL); lymphocyte-rich (LRHL); and lymphocyte-depleted (LDHL) [5].

Autologous hematopoietic CD34+ cells transplantation is an effective therapeutic strategy for HL patients [6]. CD34 is a member of a family of single-pass transmembrane sialomucin proteins that are widely expressed on small subsets of bone marrow (BM), umbilical cord blood and peripheral blood cells. Several studies have shown that CD34+ cells can affect disease progression, as a high dose of CD34+ cells is associated with better outcome in cHL [7], and the use of CD34+ cells for autografting is feasible, safe, and an effective procedure for relapsed and refractory HL [8]. Besides, CD34 expression is also found on multiple cancer stem cells [9], but not lymphoid cells [10] and is typically lost during cell maturation. A previous study indicated that the number of BM CD34^+^ cells is higher in patients with non-Hodgkin lymphoma than healthy individuals [11]. Acute lymphoblastic leukemia (ALL) cases with high expression of CD34 exhibit worse outcome [12].

There are several mechanisms, including gene mutations and aberrant activation of signaling pathways, resulting in uncontrolled and disproportional B-cell proliferation in HL [13]. The NF-κB, JAK-STAT, PI3K/AkT/GSK3β, ERK, Wnt, PD-L1/PD-L2, and NOTCH pathways have been identified to contribute to the rescue of HRS cells from apoptosis by their constitutive activations [3,14,15]. In addition, genes with high mutation frequencies in cHL are tumor necrosis factor alpha-induced protein 3 (TNFAIP3) [16], human leukocyte antigen (HLA), kinase insert domain receptor (KDR) [17], c-Rel, P53, CD95/Fas [18], protein tyrosine phosphatase non-receptor type 1 (PTPN1), G protein subunit alpha (GNA)13, and β2 microglobulin (B2M) [19]. These genes have been considered tumor suppressor genes by affecting the inflammatory response, carcinogen metabolism enzymes, and immune invasion in cHL.

In addition to the common gene mutations, other candidate genes such as NCF1 (neutrophil cytosolic factor 1), MMP9 (matrix metalloproteinase 9), and VDR (vitamin D receptor) have recently been shown to associate with the regulation of the cellular physiological processes in cHL. NCF1 is a component of NADPH oxidase 2 (NOX2) complex that is expressed in neutrophils, dendritic cells, and macrophages. The NCF1 participates in the pathogenesis of autoimmune and inflammatory diseases [20,21] and its expression is downregulated in HRS cells in cHL patients [22]. A NCF1 rs201802880 variant is linked to impaired extracellular reactive oxygen species (ROS) production and hyperactivation of the interferon (IFN) type 1 signaling [20] as well as altering plasmacytoid dendritic cell function [21]. The MMP9 is a significant protease that degrades the extracellular matrix, and its increased expression is a poor prognosis factor for cHL [23]. Vitamin D receptor (VDR) acts as a nuclear transcription factor which forms a complex with 1,25(OH)2D3 and participates in the regulation of cell biological properties. The expression level of VDR is also associated with clinical and pathologic features in cHL [24].

In this study, we aim to explore gene mutations in CD34+ cells from BM by using whole exome sequencing (WES) technology to search for new pathogenic genes in cHL. Moreover, the levels of abnormal genes present at high mutation frequencies in cHL such as CNN2 and MUC4 and their associations with clinical properties were determined.

## 2. Materials and Methods

### 2.1. Bone Marrow and Blood Sample Collection

A total of 60 untreated patients who were diagnosed with cHL from August 2021 to August 2024 and 40 healthy controls were recruited into the study at the National Institute of Hematology and Blood Transfusion, 103 Military, 108 Center Military, and Vietnam National Cancer Hospitals, Ha Noi, Vietnam. The exclusion criteria were an age of <17 years; those with comorbid autoimmune or inflammatory diseases; and a family history of lymphoma or certain genetic disorders. The diagnosis of cHL was based on the 2016 WHO criteria [5] and classified into four histological subgroups: nodular sclerosis (NSHL, *n* = 31), mixed cellularity (MCHL, *n* = 21), lymphocyte-rich (LRHL, *n* = 7), and lymphocyte-depleted cHL (LDHL, *n* = 1). No individuals in the control population took any medication or suffered from any known acute or chronic disease.

### 2.2. Cell Separation and Flow Cytometry Analysis

Bone marrow and peripheral blood mononuclear cells (PBMC) of cHL patients were isolated using Ficoll-Hypaque (GE Healthcare, Chicago, IL, USA) gradient centrifugation. Bone marrow cells were then processed for the isolation of CD34+ cells using CD34 MicroBead Kit UltraPure (Miltenyi Biotech, Auburn, CA, USA) and following manufacturer protocol. The purity of the BM CD34+ cells were greater than 95% as determined by flow cytometry (FACSAria Fusion, BD Biosciences, Franklin Lakes, NJ, USA) using anti-human CD34 APC (Cat # A16226, Thermo Fisher Scientific, Waltham, MA, USA). The viability was higher than 80% in all cases.

### 2.3. Whole Exome Sequencing

Whole exome sequencing (WES) was performed on genomic DNA of BM CD34+ cells obtained from cHL patients at diagnosis, using DNeasy Blood & Tissue Kit (Qiagen, Hilden, Germany) according to the manufacturer’s protocol. Briefly, library preparation was performed using an Illumina DNA Prep kit (Illumina Inc., San Diego, CA, USA) and sequenced on Illumina NextSeq 500 instruments (Illumina Inc.), which generated 2 × 150 bp paired-end reads. Sequencing reads were mapped and aligned to the GRCh38 human reference genome using the Burrows–Wheeler Alignment (BWA, version 0.7.17) package and followed by sorting (Samtools v1.3).

### 2.4. Variant Calling and Annotation

Variants were filtered according to coverage (≥30) and preferentially selected for further analysis and validation if they met the following criteria: (a) variants with minor allele frequency (MAF) <0.01 in the 1000 Genomes Project database (http://www.internationalgenome.org/), Genome Aggregation database (gnomAD, http://gnomad.broadinstitute.org/, accessed on 18 November 2024), Single Nucleotide Polymorphisms database (dbSNP, https://www.ncbi.nlm.nih.gov/snp, accessed on 18 November 2024), ClinVar database (http://www.ncbi.nlm.nih.gov/clinvar/, accessed on 20 December 2024), and 535 in-house Vietnamese exomes and (b) they were nonsynonymous single nucleotide polymorphisms (SNPs) or small insertion and deletion (INDEL) mutations.

Statistical analysis of SNPs and INDEL mutations were called using Genome Analysis Tool Kit (GATK, ver 4.3) software [25]. Effects of SNPs were predicted to determine whether an amino acid substitution affects protein function via the ANNOVAR version 3 [26], which used the following 6 biotools: Sorting Intolerant from Tolerant (SIFT, damaging), Polymorphism Phenotyping2 (PolyPhen2_HDIV, possibly damaging or damaging), MutationAssessor (medium or high), MutationTaster (disease causing), Functional Analysis through Hidden Markov Model (FATHMM, deleterious), and Likelihood Ratio Test (LRT, deleterious) to focus on the functional prediction of variants [27,28,29].

### 2.5. Sanger Sequencing Validation

Variants in cHL-associated genes including NCF1, MMP9, and VDR were validated by Sanger sequencing. To determine polymorphisms of the NCF1, MMP9, and VDR genes, a polymerase chain reaction (PCR) and DNA sequencing (3500 Genetic Analyzers, Thermo Scientific) were performed as previously described [30]. The GenBank accession numbers NM_000265.7, NM_004994.3, and NM_000376.3 were used for DNA sequence analysis of NCF1, MMP9, and VDR genes, respectively, by using primers: NCF1-F: 5′-ACTCCTGACCTCAAGTGATCCA-3′ and NCF1-R: 5′-CAAAACACAGAAAGTCCCACCC-3′; MMP9-F: 5′-AGTGGGCTGATACCGTCTCT-3′ and MMP9-R: 5′-CCCAGGGGCCACACATTAAA-3′; VDR-F: 5′-AGAGGTGAGAGTGACTGGCA-3′ and VDR-R: 5′-GCATCTGACCCTGGACTTCC-3′. The amplification product lengths of NCF1, MMP9, and VDR were 590, 495, and 555 bp, respectively. All obtained PCR fragments and were purified with a GeneJET PCR purification kit (Thermo Scientific). The PCR products were sequenced on both strands with the same primers used for the PCR.

### 2.6. RNA Extraction and Quantitative Real-Time PCR

Total mRNA was isolated from PBMCs using the Qiashredder and RNeasy Mini Kit from Qiagen according to the manufacturer’s instructions. For cDNA first-strand synthesis, 1 µg of total RNA in 12.5 µL of DEPC-H2O was mixed with 1 µL of oligo-dT primer (500 µg/mL, Invitrogen) and heated for 2 min at 70 °C. To determine transcript levels of CNN2, MUC4, and GAPDH, the quantitative real-time PCR with the LightCycler System (Roche Diagnostics, Basel, Switzerland) was applied. The following primers: CNN2 primers: 5′-GGTCAAGGCCATATCCCAATAC-3′ (forward) and 5′-GGCATAGAAACCACAAACTGCTC-3′ (reverse); MUC4 primers: 5′-TTCTAAGAA CCACCAGACTCAGAG C-3′ (forward) and 5′-GAGACACACCTGGAGAGAATGAGC-3′ (reverse); and GAPDH primers: 5′-GGAGCGAGATCCCTCCAAA-3′ (forward) and 5′-GGCTGTTGTCATACTTCTCAT-3′ (reverse) were used. PCR reactions were performed in a final volume of 20 µL containing 2 µL cDNA, 2.4 µL MgCl_2_ (3 µM), 1 µL primer mix (0.5 µM of both primers), 2 µL cDNA Master SybrGreen I mix (Roche Molecular Biochemicals, Basel, Switzerland), and 12.6 µL DEPC-treated water. The target DNA was amplified during 40 cycles of 95 °C for 10 s, 62 °C for 10 s, and 72 °C for 16 s, each with a temperature transition rate of 20 °C/s, a secondary target temperature of 50 °C, and a step size of 0.5 °C. Melting curve analysis was performed at 95 °C, 0 s; 60 °C, 10 s; and 95 °C, 0 s to determine the melting temperature of primer dimers and the specific PCR products. The ratio between the respective gene and corresponding GAPDH was calculated per sample according to the ∆∆ cycle threshold method [31].

### 2.7. Statistical Analysis

Statistical analysis was performed with SPSS version 28 and GraphPad Prism 8 version 8 software. Data were analyzed by using the Mann–Whitney U test. *p* < 0.05 was considered statistically significant.

## 3. Results

### 3.1. Clinical Manifestations of cHL Patients

Sixty patients with cHL were enrolled and their clinical profiles showed significant increases in globulin, ferritin, and lactic acid dehydrogenase (LDH) concentrations as well as the number of neutrophils, whereas levels of hemoglobin and hematocrit were reduced in all patients. Moreover, levels of β2 microglobulin were elevated in all patients, except for the NSHL. Levels of glucose were higher in the MCHL, while the concentrations of total protein and ALT as well as the numbers of white blood cells, lymphocytes, and eosinophils were increased in the LRHL group as compared with their cutoff values (Table 1). Moreover, levels of TGF-β were higher in cHL patients than healthy controls (Figure 1). In addition, no changes in other clinical indicators of the patient groups were found (Table 1).

### 3.2. Whole Exome Sequencing and Pathogenic Mutation Profile of cHL Patients

WES was performed on BM CD34+ cell samples from thirty patients with cHL. In total, WES revealed 39,720 ± 4830 variants with at least 30X coverage, 3.74% (1484.2 ± 123.7 SNPs) of which were rare by filtering based on MAF ≤ 0.01 (Figure 2A). Within the rare variants, 32.67 ± 6.67 were frameshift INDEL, 19.47 ± 3.51 were non-frameshift INDEL, 1.83 ± 1.23 were start-loss, 0.2 ± 0.55 were stop-loss, and 8.43 ± 2.64 were stop-gain (Figure 2B).

Pathogenic variants were implemented to predict the pathogenicity of the variants, using different software tools (by means of at least four out of the six tools), including SIFT (damaging), PolyPhen 2 HDIV (damaging), MutationTaster (disease causing), MutationAssessor (medium or high), LRT (deleterious), and FATHMM (damaging). The pathogenic variants were found not only in cHL-related genes, but also in other candidate genes, highlighting the diverse genotypes in the BM CD34+ cells of cHL cases. The detail of the 29 rare variants in 23 genes in each patient was presented in Figure 2C.

Of these variants, NCF1 [22], MMP9 [23], and VDR [24] were related to the development of cHL with carrier frequencies of 13.33%, 3.33%, and 3.33%, respectively (Table 2). The prediction of the NCF1 rs201802880 mutation by SIFT, PolyPhen 2 HDIV, LRT, MutationTaster, MutationAssessor, and FATHMM turned out to be damaging (score = 0.029), possibly damaging (rank score = 0.786), deleterious (rank score = 0.629), disease causing (score = 1), medium (score = 3.335), and deleterious (rank score = 0.866), respectively. In this study, we did Sanger sequencing validation for all three genes NCF1, MMP9, and VDR. Figure 3 confirmed the variants of the three genes, which were shown by the WES variant call.

### 3.3. Novel Genetic Alterations in cHL Not Otherwise Specified

We also found 14 variants in nine genes involved in the pathogenesis of lymphoma and/or leukemia. Among them, CNN2 is involved in the regulation of the development of follicular lymphoma [32]. The carrier frequencies of CNN2 rs77830704 and rs78386506 variants in cHL cases were 76.67% and 63.33%, respectively (Table 2). Prediction by SIFT, PolyPhen 2 HDIV, MutationTaster, MutationAssessor, and FATHMM of the rs77830704 mutation were damaging (score = 0.001), damaging (rank score = 0.991), disease causing (score = 1), medium (score = 2.12), and deleterious (rank score = 0.952); and of the rs78386506 mutation turned out to be damaging (score = 0), benign (rank score = 0.42), disease causing (score = 0.991), medium (score = 1.995), and deleterious (rank score = 0.915), respectively.

Next, the MUC4 gene is known to be mutated in B-lymphoblastic leukemia [33], and B-cell lymphoma [34]. In this study, six pathogenic variants of the MUC4 gene were carried with relatively high proportions in cHL patients (Appendix A). High frequencies of the six pathogenic variants were noted in c.9832_9927 DelCTGAGGAAGGGCTGGTGACATGAAGAGGGGTGGCGTGACCTGTGGATGCTGAGGAAGCGTCGGTGACAAGAAGAGGAGTGGCGTGACCTGTGGATA (p.Y3278_Q3209Del, 66.67%), c.3228_3371 DelGGTGGTGTGACCTGTGGATACTGAGGAAGTGTCGGTGACAGGAAGAGAGGTGGCGTGACCTGTGGATGCTGAGGAAGTGTCGGTGACAGGAAGAGGGGTGGTGTGACCTGTGGATACTGAGGAAGTGTCGGTGACAGGAAGAGA (p.P1076_P1124Del, 33.33%), rs748236754 (p.P1641Ins, 26.67%), c.4826 InsTGGTGACAGGAAGAGGGGTGGCGTGACCTGTGGATGCTGAGGAAGGGC (p.P1609Ins, 23.33%), rs748705487 (p.T4138_A4154Del, 20%), and c.12363_12410 DelGGTGACAGGAAGAGAGGTGGTGTGACCTGAGGATGCTGAGGAAGGGAT (p.P4121_P4137Del, 16.67%). Of these six potential pathogenic variants in the MUC4 gene, four were novel (Appendix A).

The next top four genes frequently observed in cHL patients were KMT2C (20%), HAVCR2 (16.67%), TCF19 (13.33%), and DUOX2 (10%) (Table 2). In addition, the carrier frequencies of ADAM12 and GSS genes were 6.67%, while one patient was carrying the G6PD gene variant (Table 2). Besides lymphoma- and/or leukemia-related genes, the most frequently mutated tumor suppressor genes included AFAP1L1, FBN3, MTSS2, and TTLL5, with each being mutated in at least 10% of cHL cases (Appendix A).

Importantly, the percentage of LRHL patients carrying the NCF1 rs77759698 variant was 75% and that of MCHL patients carrying the ADAM12 rs144561426 and TTLL5 rs375900619 were 100% and 66.67%, respectively. The KMT2C rs201834857 and the FBN3 rs540022951 variants were carried by NSHL patients with percentages of 60% and 66.67%, respectively.

### 3.4. Association Between CNN2 and MUC4 Expression Levels and Clinical Features in cHL

Last but not least, to ask whether the high-frequency mutation genes, such as CNN2 and MUC4 are involved in epigenetic regulation, their mRNA expression levels were further evaluated by quantitative real-time PCR. Gene expression profiling revealed that the mRNA expression levels of CNN2 and MUC4 were significantly increased in cHL patients (Figure 4).

The association between CNN2 and MUC4 expression levels and clinical features was also determined. The levels of CNN2 and MUC4 were divided into two groups based on the median CNN2 and MUC4 expression values in healthy controls (high vs. low). The high CNN2 expression group was detected in 42/60 samples (70%) and the low CNN2 expression group was detected in 18/60 samples (30%). The high MUC4 expression group was detected in 34/60 samples (56.67%) and the low MUC4 expression group was detected in 26/60 samples (43.33%) (Appendix A). Results indicated that neutrophil-to-lymphocyte ratio (NLR) was lower in the high MUC4 expression than that in the low MUC4 expression group. Unlike MUC4, the patients with the high CNN2 expression had lower levels of LDH and higher levels of hematocrit as compared to the low CNN2 expression group (Appendix A). In addition, the significant differences in other clinical indicators among the groups did not exist.

## 4. Discussion

Unlike most other malignant tumors, cHL is not characterized by a high number of proliferating tumor cells, and the HRS tumor cells represent only less than 2% of the total cells. A recent study indicates that high dose of CD34+ cells improves outcome and overall survival rate in cHL [7], suggesting that these cells were functionally defective and their gene mutations may occur in cHL patients. Another study indicates the efficacy of autologous stem cell transplantation in relapsed or refractory HL [8]. CD34 is well known to highly express on a variety of cancer stem cells, but not in mature blood and lymphoid cells [9,10] and plays an important role in blocking cell differentiation [9]. A recent study reveals that CD34^+^CD19^+^ cells are indicated as candidate lymphoma stem cells (LSC) and follicular lymphoma patients have a higher percentage of these cells in BM [35]. Their genetic alterations are also observed in these patients [36]; therefore, we ask whether gene mutations are present in BM CD34+ cells from cHL patients.

In this study, WES analysis was conducted on BM CD34+ cells from thirty patients with cHL. Results showed that carriers of cHL-related gene variants, including NCF1, MMP9, and VDR were at percentages of 13.33%, 3.33%, and 3.33%, respectively. These genes are known to be involved in abnormal cHL pathogenesis and described as epigenetic regulators of cHL. Attenuated expression of the NCF1 is found in HRS cells of cHL patients [22]. The NCF1 rs201802880 mutation alters function of plasmacytoid dendritic cells [21], which is associated with hyperactivation of the interferon (IFN) type 1 signaling [20]. MMP-9 is produced by lymphocytes and macrophages and plays a crucial role in the regulation of the pathogenesis of multiple cancers [34]. The increased expression of MMP-9 is a poor prognosis factor for cHL [21] by promoting cancer cell migration [37]. Similar to NCF1 and MMP-9, VDR is linked to modulation of immune cell function [38] and considered a specific marker for HL, but not of B-cell derived non-Hodgkin lymphoma [24].

Importantly, we observed that other candidate pathogenic variants, including CNN2 rs77830704 (76.67%), CNN2 rs78386506 (63.33%), MUC4 p.Y3278_Q3209Del (66.67%), MUC4 p.P1076_P1124Del (33.33%), MUC4 rs748236754 (26.67%), MUC4 p.P1609Ins (23.33%), MUC4 rs748705487 (20%), MUC4 p.P4121_P4137Del (16.67%), and MTSS2 rs531163149 were found with high frequencies in this cohort. Among the three genes, the CNN2 and MUC4 genes are known to function as regulators of cell physiological processes, including differentiation, proliferation, adhesion, apoptosis, and inflammatory response [39,40]. CNN2 enhances the development of follicular lymphoma cells [32] by inducing their migration and invasion. Although MUC4 is noticed to be recurrently mutated in B-lymphoblastic leukemia [33], and B-cell lymphoma [34], the six pathogenic variants in MUC4 were not overlapped, in which four mutations were novel. MUC4 overexpression confers tumor cells to apoptotic resistance [41].

More importantly, the levels of MUC4 were significantly associated with the development of lymphocytes and neutrophils in cHL, as the NLR was lower in the high MUC4 expression compared to the low MUC4 expression group. An increase in NLR is associated with poor prognosis and adversely affects survival in cHL [42]. Unlike MUC4, the low CNN2 expression group had higher levels of LDH and lower levels of hematocrit. The low hematocrit and high LDH levels are also poor prognosis features in cHL [43]. Therefore, the low expressions of CNN2 and MUC4 might be important risk factors for poor prognosis in cHL.

In addition, the KMT2C rs201834857, HAVCR2 rs184868814, and TCF19 rs541001159 SNPs were detected with relatively high frequencies in at least 13.33% of cHL cases. The roles of the three genes in regulating functions of lymphoma cells have been recently reported. The KMT2C gene is critical for enhancer activation, cell differentiation, and development and its mutation occurs at a frequency of 19% in patients with mucosa-associated lymphoid tissue (MALT) lymphoma [44]. The germline HAVCR2Y82C mutation results in persistent immune activation [45] and is harbored by 79.4% of patients with subcutaneous panniculitis-like T-cell lymphoma [46]. Similarly, a rs7750641 SNP in the TCF19 gene is a risk variant of MALT lymphoma [47]. In addition, TCF19 promotes proliferative and migratory capacities of cancer cells [48].

Unlike several studies that found the most frequently mutated genes in cHL were TNFAIP3, HLA, KDR, c-Rel, P53, CD95/Fas, PTPN1, GNA13, and B2M [16,17,18,19], which are involved in aberrant activation of signaling pathways promoting cell proliferation and survival, as well as immune evasion [3,14]. In this study, we focused on investigating gene mutations in BM CD34+ cells, which account for 1–2% of the total BM cells in healthy individuals [8] and about 2–3% in cHL patients, therefore, the common gene mutations were absent. This finding revealed for the first time the possible pathogenic variants in BM CD34+ cells of cHL patients.

In addition, levels of TGF-β, which were released by HRS cells [2], are higher in the patient group than healthy controls, suggesting a hyperactive immune response in cHL cases [49]. However, no significant difference was found among the patient groups.

## 5. Conclusions

The results showed the pathogenic variants in CNN2 and MUC4 genes as the most frequent genetic alterations as well as their abnormal expressions in cHL patients. WES revealed significantly mutated genes, including several known (NCF1, MMP9, and VDR) and possibly other candidate (CNN2, MUC4, MTSS2, KMT2C, HAVCR2, and TCF19) genes, most of which were associated with the physiological activation of lymphoma cells. Moreover, the low CNN2 and MUC4 levels might be important risk factors for poor prognosis in cHL. This finding contributed to the identification of novel genotypes that might impact on function of BM CD34+ cells in cHL patients.

## Figures and Tables

**Figure 1 cimb-47-00880-f001:**
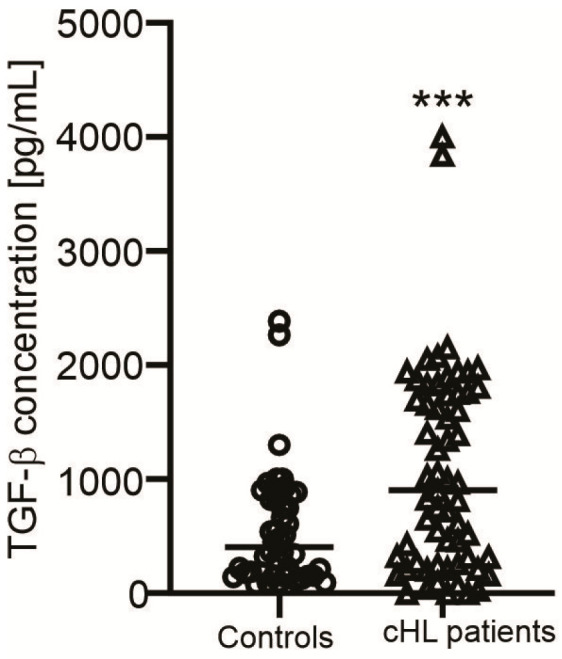
TGF-β concentrations in cHL patients. The graph indicates TGF-β concentrations in cHL patients and healthy individuals and each dot represents a single sample. *** (*p* < 0.001) shows significant difference from healthy individuals (Mann–Whitney U test).

**Figure 2 cimb-47-00880-f002:**
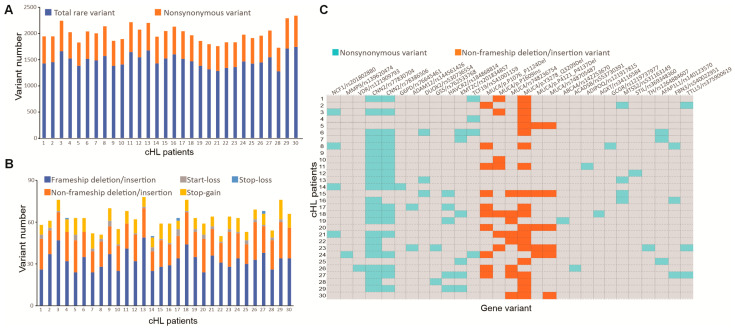
Overview of the WES analysis. (**A**,**B**). The bar graphs indicate the variant numbers in each patient. (**C**). The 29 rare variants in 23 genes of 30 cHL patients. Each row represents a patient and each column represents a gene.

**Figure 3 cimb-47-00880-f003:**
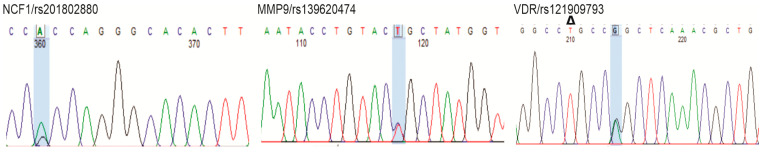
DNA sequencing chromatograms of NCF1, MMP9, and VDR genes in cHL patients.

**Figure 4 cimb-47-00880-f004:**
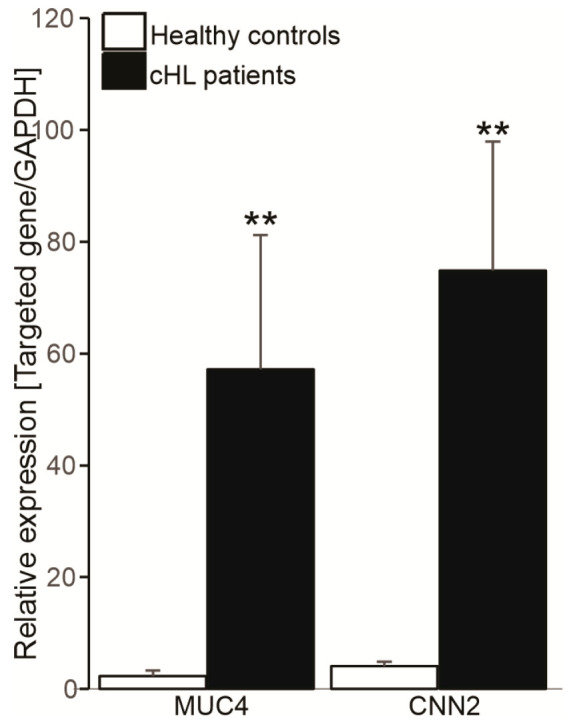
The expression levels of CNN2 and MUC4 in cHL patients. GAPDH was used as a reference gene for relative quantification. ** (*p* < 0.01) shows significant difference from healthy individuals (Mann–Whitney U test).

**Table 1 cimb-47-00880-t001:** Clinical characteristics of the study population at diagnosis.

Characteristics	Normal Value	NSHL (*n* = 31)	MCHL (*n* = 21)	LRHL (*n* = 7)	LDHL (*n* = 1)
Age (years)		31.17 ± 10.05	33.14 ± 15.04	52 ± 20.45	76
Sex, Female (n. %)		13 (40.63)	12 (54.55)	1 (14.29)	0
Urea (mmol/L)	3.3–6.6	4.67 ± 1.59	4.5 ± 1.11	5.3 ± 1.37	4.6
Glucose (mmol/L)	3.9–5.6	5.1 ± 1.48	4.78 ± 1.05	4.74 ± 0.93	6.7
Creatinine (µmol/L)	50–110	72.64 ± 15.1	71.82 ± 12.57	82.28 ± 19.3	73
Uric acid (µmol/L)	< 420	338.39 ± 76.7	327.03 ± 76.96	303.87 ± 96.48	139
Total bilirubin (µmol/L)	0–21	9.45 ± 6.57	9.79 ± 3.69	8.83 ± 4.78	153.8
Direct bilirubin (µmol/L)	0–7	2.3 ± 2.36	2.88 ± 1.84	3.28 ± 2.76	88.7
Indirect bilirubin (µmol/L)	≤12.7	7.02 ± 2.16	7.4 ± 3.26	7.46 ± 3.04	65.1
Total protein (g/L)	60–80	78.62 ± 6.92	80.1 ± 6.87	85.08 ± 12.59	58.3
Albumin (g/L)	35–52	39.3 ± 4.73	39.16 ± 5.57	38.07 ± 7.7	33.6
Globulin (g/L)	20–35	39.8 ± 6.97	41.89 ± 8.56	56.7 ± 23.19	24.7
Ferritin (ng/mL)	10–300	598.26 ± 579.89	792.19 ± 794.59	888.07 ± 585.78	2.4
AST (GOT) (U/L)	5–40	23.7 ± 13.42	23.09 ± 14.49	30.13 ± 15.92	68
ALT (GPT) (U/L)	7–55	34.25 ± 36.06	26.11 ± 23.64	62.88 ± 49.5	55
LDH (U/L)	0–247	325.75 ± 147.53	369.32 ± 289.88	337.89 ± 268.22	297
β2 microglobulin (mg/L)	0.8–2.4	1.97 ± 0.72	2.88 ± 1.63	4.52 ± 3.64	6.02
Erythrocytes (10^12^ cells/L)	4.2–5.9	4.74 ± 0.52	4.55 ± 0.95	4.4 ± 0.84	2.31
Hemoglobin (g/L)	130–180	127.77 ± 23.38	122.10 ± 23.91	127 ± 23.38	75
Hematocrit (%)	42–52	40.4 ± 3.85	37.8 ± 6.65	39 ± 7.24	22.9
WBC count (×10^9^/L)	5–14.5	12.16 ± 4.77	11.63 ± 5.8	14.7 ± 6.6	1.31
Neutrophil count (×10^9^/L)	1.6–7.5	9.37 ± 4.52	8.15 ± 4.96	8.89 ± 6.82	0.73
Lymphocyte count (×10^9^/L)	0.9–3.4	1.67 ± 0.56	1.67 ± 0.88	3.95 ± 5.39	0.432
Monocyte count (×10^9^/L)	0–1.2	0.69 ± 0.35	0.89 ± 0.36	0.7 ± 0.18	0.144
Eosinophil count (×10^9^/L)	0–0.8	0.29 ± 0.22	0.29 ± 0.29	1.32 ± 0.36	0
Basophil count (×10^9^/L)	0–0.3	0.07 ± 0.07	0.07 ± 0.076	0.12 ± 0.15	0
Platelet count (×10^9^/L)	150–400	364.84 ± 97.21	352.62 ± 124	310.29 ± 127.88	107

ALT, alanine aminotransferase; AST, aspartate aminotransferase; LDH, lactate dehydrogenase; NS, nodular sclerosis; MC, mixed cellularity; LD, lymphocyte-depleted; LR, lymphocyte-rich; and WBC, white blood cells.

**Table 2 cimb-47-00880-t002:** Genotype distribution of gene variants in cHL patients.

**Gene Name**	**dbSNP ID**	**Type of Variant**	**Transcription**	**Locus**	**Nucleotide Change**	**Amino Acid Change**	**Variant Allele Frequency (%)**	Scores of SIFT/PolyPhen2/LRT/MutationTaster/MutationAssessor/FATHMM	SIFT/PolyPhen2/LRT/MutationTaster/MutationAssessor/FATHMM	ExAC/1000 Genome (Frequency)	ClinVar	Alpha Fold Database (Model Confidence)
**Hodgkin lymphoma-associated gene**												
*NCF1*	rs201802880	Missense	NM_000265.7	Exon 4	c.269G>A	p.R90H	13.33	0.029/0.786/0.629/1/3.335/0.866	D/P/D/D/M/D	0.0011/0	USV	Very high
*MMP9*	rs139620474	Missense	NM_004994.3	Exon 2	c.150C>T	p.R51C	3.33	0.002/1/0.523/1/4.025/0.417	D/D/D/D/H/T	0.000019/0	Likely pathogenic	Very high
*VDR*	rs121909793	Missense	NM_000376.3	Exon 4	c.239G>A	p.R80Q	3.33	0.012/0.996/0.629/1/4.895/0.985	D/D/D/A/H/D	0.00004/0	Pathogenic	Very high
**Lymphoma- and/or leukemia-related gene**												
*CNN2*	rs77830704	Missense	NM_004368.4	Exon 7	c.787G>A	p.G263S	76.67	0.001/0.991/0.843/1/2.12/0.952	D/D/D/D/M/D	0.0001/0	Pathogenic	Low
*CNN2*	rs78386506	Missense	NM_004368.4	Exon 7	c.797G>A	p.R266Q	63.33	0/0.42/0.627/0.991/1.995/0.915	D/B/D/D/M/D	0.000085/0	Pathogenic	Low
*G6PD*	rs76645461	Missense	NM_001360016.2	Exon 3	c.143T>C	p.I48T	3.33	1/0.985/0.843/1/1.365/0.982	T/D/D/D/L/D	0.000011/0.000265	Likely pathogenic	Very high
*ADAM12*	rs144561426	Missense	NM_001288973.2	Exon 9	c.854T>C	pD285G	6.67	0.054/1/0.843/1/1.975/0.634	T/D/D/D/M/T	0.0001/0	USV	Very high
*DUOX2*	rs530736554	Missense	NM_001363711.2	Exon 12	c.1295G>A	p.R432H	10	0.038/0.933/0.629/1/2.75/0.842	D/P/D/D/M/D	0.0000676/0.000998	USV	Very high
*GSS*	rs762533768	Missense	NM_000178.4	Exon 5	c.383A>G	p.Y128C	6.67	0.001/1/0.629/1/3.74/0.96	D/D/D/D/H/D	0.000004/0	USV	Very high
*HAVCR2*	rs184868814	Missense	NM_032782.5	Exon 2	c.245A>G	p.Y82C	16.67	0/1/0.505/1/3.765/0.668	D/D/D/D/H/T	0.0038/0.00619	USV	High
*KMT2C*	rs201834857	Missense	NM_170606.3	Exon 8	c.1042G>A	p.D348N	20	0.027/1/0.438/1/2.62/0.988	D/D/U/D/M/D	0.0003/0	USV	High
*TCF19*	rs541001159	Missense	NM_007109.2	Exon 3	c.755A>G	p.R252G	13.33	0.009/0.998/0.629/0.626/2.47/0.278	D/D/D/D/M/T	0.0001/0.000799	USV	Very high

SIFT, “D” meaning damaging, score < 0.05, “T” meaning tolerated, score ≥ 0.05; PolyPhen2, “D” meaning damaging, 0.957 ≤ score ≤ 1, “P” meaning possibly damaging, 0.453 ≤ score ≤ 0.956, “B” meaning benign, 0 ≤ score ≤ 0.452; LRT, “D” meaning deleterious, “U” meaning unknown; MutationTaster, “A” and “D” represent as disease causing automatic and deleterious, 0.84 ≤ score ≤ 1; MutationAssessor, “H” meaning high impact, score > 3.5, “M” meaning medium impact, 1.9 ≤ score ≤ 3.5, "L" for the MutationAssessor tool is "low"; FATHMM, “D” meaning damaging and “T” meaning tolerated, score > 0.5.

## Data Availability

The datasets used and/or analyzed during the current study are available from the corresponding author on reasonable request.

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
