# Peer review of "Whole Exome Sequencing for the Identification of Mutations in Bone Marrow CD34+Cells in Hodgkin Lymphoma"

_cimb, 2025, doi:10.3390/cimb47110880_

Round 1
Reviewer 1 Report
Comments and Suggestions for Authors
Summary :
In the presented manuscript titled “Whole-exome sequencing for the identification of mutations in bone marrow CD34+cells in Hodgkin lymphoma”, Trang et al. report whole-exome sequencing (WES) in CD34+ bone marrow cells of Hodgkin lymphoma patients (n=30). The authors report variants in NCF1, MMP9 (3.33%), and VDR (3.33%), as well as numerous other variants across different genomic loci (MUC4, CNN2, MTSS2, KMT2C). Three variants were confirmed with Sanger Sequencing. The authors next assessed RNA expression of candidate loci CNN2 and MUC4 and report widespread overexpression of these two genes in PBMCs from Hodgkin Lymphoma patients. They link increased CNN2 and MUC4 levels to distinct clinical blood markers, namely neutrophil-to-lymphocyte ratio (NLR) for MUC4 and LDH/hematocrit for CNN2. The authors conclude that CNN2 and MUC4 have emerged as new loci associated with Hodgkin lymphoma via the presence of gene SNPs in CD34+ cells.
General concept comments:
The study highlights a complex picture of rare SNPs in CD34+ cells of Hodgkin Lymphoma patients. The findings and methodology are presented mostly clear and in a logical order. The identified variants are explored via in silico tools and gene expression analysis to assess their potential biologic impact, thus strengthening the study’s findings.
One major drawback of the study is its limited statistical power, with WES from only 30 patients reported. Of note, many candidate SNPs presented (e.g. MMP9 and VDR) were detected only in one single patient, making it statistically unclear how common these SNPs really are. However, the authors mitigate this concern by focusing on more common SNPs in the MUC4 and CNN2 genes. These SNPs are commonly found in the study cohort, but carry a MAF of <0.01 in control databases. Overall, the study is meritorious in identifying novel MUC4 SNPs, and by associating SNPs in CD34+ cells to Hodgkin lymphoma. It provides a novel approach to the understanding of Hodgkin lymphoma, and provides compelling arguments for the biological roles of the identified SNP loci. The authors’ conclusions are supported by their findings and are discussed in the context of the published literature in a meaningful way.
However, there are several aspects of the study that can be improved or need further clarification. This is especially true for the selection criteria that led to the identification of candidate SNPs (table 2). These were classified as Hodgkin lymphoma associated, leukemia/lymphoma-associated or cancer associated, but no details are given how that association was established and a statistical approach to how this shortlist of 29 SNPs was selected is not clear. The authors need to provide a clearer rationale for choosing these SNPs and support that via a more detailed description of selection criteria (e.g. via adding literature references to table 2, or via adding more detail to the methods section). In addition, other details regarding the study results and methods are unclear, and should be described in more detail, as outlined below in the specific comments.
Specific comments:
- Line 64 and line 300 “treatment potential”: It is unclear what is meant by this statement. The cited paper states that lower number of transfused CD34+ cells was associated with significantly lower overall survival. The authors should rephrase the statement and be more precise about the “treatment potential” they are referring to.
- Line 117 “The purity of the BM CD34+ cells were greater than 95%”: The authors should mention specifically the antibody (with manufacturer and catalog-number/unique ID) that was used for flow-cytometric CD34 assessment in the cells.
- Line 122 “library preparation”: The authors should state specifically which method or kit they used for WES library preparation.
- Line 193 “were increased as compared their cutoff values in the LRHL group”: The authors should indicate statistical significance of these comparisons in table 1. In its current form, the table only shows averages and standard deviations, but does not indicate which differences are statistically significant.
- Line 206-211 “WES revealed 39.720 ± 4830 variants with at least 30X coverage, 3,74% (1484,2±123,7 SNPs) […] 8,43±2,64 were stop-gain”: Commas (,) and dots (.) were seemingly swapped in the presentation of numbers here compared to the rest of the manuscript. For example, the number of variants should likely read “39,720” (thirty nine thousand seven hundred twenty), and the number of rare variants should read “1484.2” (one thousand four hundred eighty four point two). The authors should double-check and harmonize use of dots and commas throughout the manuscript, with dots being used for decimals.
- Line 236-242 “Prediction by SIFT, 236 PolyPhen 2 […] deleterious (rank score = 241 0.915), respectively”: The authors present in silico predictions of the potential impact of nonsynonymous SNPs on protein function. Since many of the presented SNPs affect protein sequence (table 2), the authors should add scores from the prediction tools they used into table 2 so that a prediction for all candidate SNPs is laid out clearly.
Some grammatical errors are still present in the manuscript (examples below). The authors are encouraged to proof-read the text and implement changes:
- Line 60 “CD34 a member” should be corrected to “CD34 is a member”
- Line 62 “peripheral blood cells Several studies […]” should be corrected to “peripheral blood cells. Several studies […]”
- Line 271-273: “to ask whether the high-frequency mutation genes, such as CNN2 and MUC4 are involved in the epigenetic regulation. Their mRNA expression levels were further evaluated by quantitative real-time PCR.”: These two sentences should be combined to make their logical connection clear: “to ask whether the high-frequency mutation genes, such as CNN2 and MUC4 are involved in the epigenetic regulation, their mRNA expression levels were further evaluated by quantitative real-time PCR.”
- Line 312 “known to involve in” should be corrected to “known to be involved in”
Author Response
There are several aspects of the study that can be improved or need further clarification. This is especially true for the selection criteria that led to the identification of candidate SNPs (table 2). These were classified as Hodgkin lymphoma associated, leukemia/lymphoma-associated or cancer associated, but no details are given how that association was established and a statistical approach to how this shortlist of 29 SNPs was selected is not clear. The authors need to provide a clearer rationale for choosing these SNPs and support that via a more detailed description of selection criteria (e.g. via adding literature references to table 2, or via adding more detail to the methods section).
Response: The selection criteria of the SNPs is described in the Method and Results sections as belows.
Variants were filtered according to coverage (≥30) and preferentially selected for further analysis and validation if they met the following criteria: (a) Variants with minor allele frequency (MAF) <0.01 in the 1,000 Genomes Project database (http://www.internationalgenome.org/), Genome Aggregation database (gnomAD, http://gnomad.broadinstitute.org/), Single Nucleotide Polymorphisms database (dbSNP, https://www.ncbi.nlm.nih.gov/snp), ClinVar database (http://www.ncbi.nlm.nih.gov/clinvar/), and 535 in‐house Vietnamese exomes; and (b) they were nonsynonymous single nucleotide polymorphisms (SNPs) or small insertion and deletion (INDEL) mutations
Pathogenic variants were implemented to predict the pathogenicity of the variants, using different software tools (means of at least four out of the six tools), including SIFT (damaging), PolyPhen 2 HDIV (damaging), MutationTaster (disease causing), MutationAssessor (medium or high), LRT (deletious), and FATHMM (damaging).
In addition, other details regarding the study results and methods are unclear, and should be described in more detail, as outlined below in the specific comments.
Specific comments:
- Line 64 and line 300 “treatment potential”: It is unclear what is meant by this statement. The cited paper states that lower number of transfused CD34+ cells was associated with significantly lower overall survival. The authors should rephrase the statement and be more precise about the “treatment potential” they are referring to.
Response: Yes, we rephrased the “treatment potential” as suggested.
- Line 117 “The purity of the BM CD34+ cells were greater than 95%”: The authors should mention specifically the antibody (with manufacturer and catalog-number/unique ID) that was used for flow-cytometric CD34 assessment in the cells.
Response: Yes, the manufacturer and catalog number of CD34 antibody is added in the Method section
- Line 122 “library preparation”: The authors should state specifically which method or kit they used for WES library preparation.
Response: Library preparation was performed using an Illumina DNA Prep kit
- Line 193 “were increased as compared their cutoff values in the LRHL group”: The authors should indicate statistical significance of these comparisons in table 1. In its current form, the table only shows averages and standard deviations, but does not indicate which differences are statistically significant
Response: Yes, they compared clinical features in the patient groups with their cutoff values in Table 1, therefore we cannot calculate statistical differences. The sentence “… were increased as compared their cutoff values in the LRHL group” is changed into “… were increased in the LRHL group as compared their cutoff values”. Thank you for your kind understanding.
- Line 206-211 “WES revealed 39.720 ± 4830 variants with at least 30X coverage, 3,74% (1484,2±123,7 SNPs) […] 8,43±2,64 were stop-gain”: Commas (,) and dots (.) were seemingly swapped in the presentation of numbers here compared to the rest of the manuscript. For example, the number of variants should likely read “39,720” (thirty nine thousand seven hundred twenty), and the number of rare variants should read “1484.2” (one thousand four hundred eighty four point two). The authors should double-check and harmonize use of dots and commas throughout the manuscript, with dots being used for decimals.
Response: Yes, they are corrected
- Line 236-242 “Prediction by SIFT, 236 PolyPhen 2 […] deleterious (rank score = 241 0.915), respectively”: The authors present in silico predictions of the potential impact of nonsynonymous SNPs on protein function. Since many of the presented SNPs affect protein sequence (table 2), the authors should add scores from the prediction tools they used into table 2 so that a prediction for all candidate SNPs is laid out clearly.
Response: We added scores from the prediction tools for the candidate SNPs
Comments on the Quality of English Language
Some grammatical errors are still present in the manuscript (examples below). The authors are encouraged to proof-read the text and implement changes:
Response: Yes, the Quality of English Language is now improved
- Line 60 “CD34 a member” should be corrected to “CD34 is a member”
Response: Yes, it is corrected
- Line 62 “peripheral blood cells Several studies […]” should be corrected to “peripheral blood cells. Several studies […]”
Response: Yes, it is corrected
- Line 271-273: “to ask whether the high-frequency mutation genes, such as CNN2 and MUC4 are involved in the epigenetic regulation. Their mRNA expression levels were further evaluated by quantitative real-time PCR.”: These two sentences should be combined to make their logical connection clear: “to ask whether the high-frequency mutation genes, such as CNN2 and MUC4 are involved in the epigenetic regulation, their mRNA expression levels were further evaluated by quantitative real-time PCR.”
Response: Yes, they are corrected
- Line 312 “known to involve in” should be corrected to “known to be involved in”
Response: Yes, it is corrected
Reviewer 2 Report
Comments and Suggestions for Authors
This manuscript applies whole-exome sequencing to identify mutations in CD34⁺ cells. The experiments are carefully conducted, and the data analysis is thorough. I have the following suggestions to improve the manuscript further:
Major comments:
- In addition to the six in silico analysis tools already used, have the authors considered applying AlphaFold to predict structural changes of the identified mutants? This could help explain their effects at the molecular level.
Minor comments:
- Line 62: A period is missing before “Several.”
- Significant figures are incorrectly represented in all tables. Please revise for consistency.
- Table 2: An extra period appears in the nucleotide changes of NVF1.
- Please carefully revise the manuscript and correct many more small typos as above.
Author Response
This manuscript applies whole-exome sequencing to identify mutations in CD34⁺ cells. The experiments are carefully conducted, and the data analysis is thorough. I have the following suggestions to improve the manuscript further:
Major comments:
- In addition to the six in silico analysis tools already used, have the authors considered applying AlphaFold to predict structural changes of the identified mutants? This could help explain their effects at the molecular level.
Response: The AlphaFold Protein Structure database for predicting structural changes of the identified mutants is shown in Table 2
Minor comments:
- Line 62: A period is missing before “Several.”
Response: Yes, it is corrected
- Significant figures are incorrectly represented in all tables. Please revise for consistency.
Response: Yes, they are corrected
- Table 2: An extra period appears in the nucleotide changes of NVF1.
Response: Yes, it is corrected
- Please carefully revise the manuscript and correct many more small typos as above.
Response: Yes, we carefully revised the manuscript
Round 2
Reviewer 1 Report
Comments and Suggestions for Authors
Summary :
In the presented revised manuscript titled “Whole-exome sequencing for the identification of mutations in bone marrow CD34+cells in Hodgkin lymphoma”, Trang et al. report whole-exome sequencing (WES) in CD34+ bone marrow cells of Hodgkin lymphoma patients (n=30). The authors report variants in NCF1, MMP9 (3.33%), and VDR (3.33%), as well as numerous other variants across different genomic loci (MUC4, CNN2, MTSS2, KMT2C). Three variants were confirmed with Sanger Sequencing. The authors next assessed RNA expression of candidate loci CNN2 and MUC4 and report widespread overexpression of these two genes in PBMCs from Hodgkin Lymphoma patients. They link increased CNN2 and MUC4 levels to distinct clinical blood markers, namely neutrophil-to-lymphocyte ratio (NLR) for MUC4 and LDH/hematocrit for CNN2. The authors conclude that CNN2 and MUC4 have emerged as new loci associated with Hodgkin lymphoma via the presence of gene SNPs in CD34+ cells.
General concept comments:
The study highlights a complex picture of rare SNPs in CD34+ cells of Hodgkin Lymphoma patients. The findings and methodology are presented mostly clear and in a logical order. The identified variants are explored via in silico tools and gene expression analysis to assess their potential biologic impact, thus strengthening the study’s findings.
The authors focus on SNPs in two gene loci (MUC4 and CNN2 genes). These SNPs are commonly found in the study cohort but carry a MAF of <0.01 in control databases. Overall, the study is meritorious in identifying novel MUC4 SNPs, and by associating SNPs in CD34+ cells to Hodgkin lymphoma. It provides a novel approach to the understanding of Hodgkin lymphoma with compelling arguments for the biological roles of the identified SNP loci. The authors’ conclusions are supported by their findings and are discussed in the context of the published literature in a meaningful way.
In the revised manuscript, the authors have addressed several reviewer comments in detail and added further clarifications to their methods- and results sections. Most notably, the 29 candidate Hodgkin Lymphoma-associated SNPs are now presented in more detail in table 2. The SNP impact scores from SIFT, PolyPhen 2, HDIV, MutationTaster, MutationAssessor and FATHMM are presented clearly in that table. Two additional databases were implemented for SNP assessment (ClinVar and Alphafold, table 2). These tools further underline the potential functional and/or pathogenic impact of several amino-acid coding SNPs.
In addition, details about the study methodology have been clarified. This includes information on the antibody used for CD34+ detection (line 118/119) and WES library preparation kit (line 124). Minor changes were implemented throughout the manuscript to address clarifications regarding comparisons of clinical characteristics in table 1 (line 196) and grammar/spelling (lines 60, 62, 283/284, 325).
Overall, the reviewer comments were addressed in full. The changes in the revised manuscript have increased transparency of the presented findings and the methodology behind the study.
Author Response
Thank you for your comments and kind support.
Reviewer 2 Report
Comments and Suggestions for Authors
Thank you for addressing my comments. I have no further concern.
Author Response

(The authors gave the same response as above.)
